# SEMANTICS-ADAPTIVE ACTIVATION INTERVENTION FOR LLMS VIA DYNAMIC STEERING VECTORS

**Weixuan Wang**[1]    **Jingyuan Yang**[2]    **Wei Peng**[3]

[1]School of Informatics, University of Edinburgh    [2]Huawei Technologies Co., Ltd.
[3]School of Engineering, RMIT University
`weixuan.wang@ed.ac.uk`   `yangjingyuan2@huawei.com`   `wei.peng3@rmit.edu.au`

## ABSTRACT

Large language models (LLMs) have achieved remarkable performance across many tasks, yet aligning them with desired behaviors remains challenging. Activation intervention has emerged as an effective and economical method to modify the behavior of LLMs. Despite considerable interest in this area, current intervention methods exclusively employ a fixed steering vector to modify model activations, lacking adaptability to diverse input semantics. To address this limitation, we propose **Semantics-Adaptive Dynamic Intervention (SADI)**, a novel method that constructs a dynamic steering vector to intervene model activations at inference time. More specifically, SADI utilizes activation differences in contrastive pairs to precisely identify critical elements of an LLM (i.e., attention heads, hidden states, and neurons) for targeted intervention. During inference, SADI dynamically steers model behavior by scaling element-wise activations based on the directions of input semantics. Experimental results show that SADI outperforms established baselines by substantial margins, improving task performance without training. SADI's cost-effectiveness and generalizability across various LLM backbones and tasks highlight its potential as a versatile alignment technique.[1]

## 1 INTRODUCTION

Large language models (LLMs) have demonstrated remarkable capabilities across many tasks (OpenAI, 2023; Touvron et al., 2023; Anil et al., 2023b;a; Mesnard et al., 2024). Nevertheless, aligning these models to target behaviors remains challenging (Longpre et al., 2023; Ding et al., 2023). Existing approaches like supervised fine-tuning (Wei et al., 2022) (SFT), Reinforcement Learning from Human Feedback (Bai et al., 2022) (RLHF), and prompt engineering (Shin et al., 2020; Wang et al., 2023) are effective but have limitations. They often require extensive datasets, struggle to prevent hallucinations, and sometimes fail to produce the desired results.

Recently, advancements in model alignment techniques, known as "activation engineering", aim to address these limitations (Subramani et al., 2022; Hernandez et al., 2023; Zou et al., 2023; Li et al., 2023b; Chen et al., 2024). Activation engineering involves making targeted modifications to the internal activations of LLMs to guide their outputs more precisely. This technique constructs steering vectors that, when integrated into the forward pass of a frozen LLM, induce specific desirable changes in the output text. However, traditional steering vectors are static and may not adapt well to the diverse semantic contexts encountered during inference (Turner et al., 2023; Rimsky et al., 2023). This misalignment between the direction of steering vector and the input's semantic direction can adversely impact the model's predictive performance, particularly when the discrepancy is substantial. These limitations highlight the need for dynamic and adaptive steering mechanisms capable of effectively handling varied input semantics.

In this work, we introduce the **Semantics-Adaptive Dynamic Intervention** (**SADI**), a novel approach designed to overcome the limitations of fixed steering mechanisms. SADI adjusts model activations by dynamically generating a steering vector tailored to each input's semantic context. Specifically, SADI utilizes activation differences from contrastive pairs to create a binary mask that

---

[1]https://github.com/weixuan-wang123/SADI

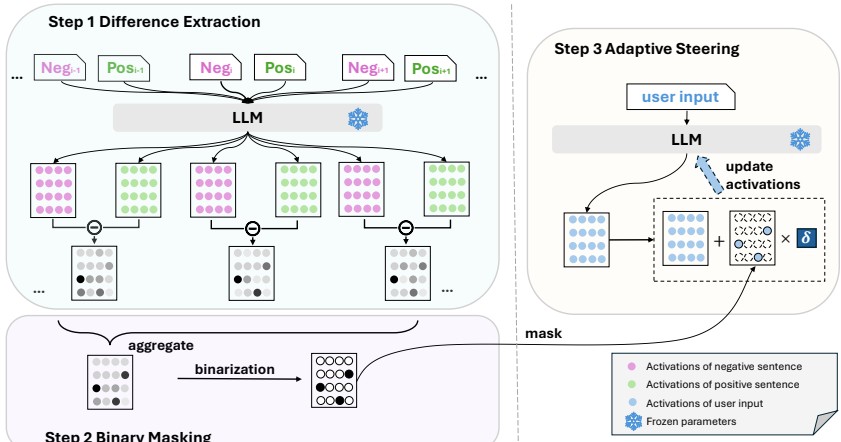

Figure 1: Three steps of SADI: (1) Difference Extraction: extract the activation differences between positive and negative examples from all model layers; (2) Binary Masking: compute the mean activation difference to locate the key elements and produce an identification mask by binarization; and (3) Adaptive Steering: intervene the activations during inference by applying the identification mask to the input activations scaled by a factor of $\delta$.

identifies critical model elements for targeted intervention. During inference, this mask is applied to user input activations with element-wise scaling, effectively manipulating the LLM's behavior to align with the input semantics. This process ensures that modifications preserve the semantic alignment of the inputs, allowing for more precise and context-sensitive interventions. Furthermore, we apply SADI to various components of LLMs, including hidden states (SADI-HIDDEN), attention heads (SADI-HEAD), and neurons in feed-forward networks (FFNs) (SADI-NEURON).

To validate the effectiveness of SADI, we conduct extensive experiments using four diverse model backbones: LLAMA2-7B-CHAT, BLOOMZ-7B, MISTRAL-7B, FALCON-7B-INSTRUCT across eleven widely used benchmarks. The experiments involve a comprehensive range of tasks, from multiple-choice tasks (COPA, StoryCloze, NLI, MMLU, SST2, SST5, BoolQ, Winogrande), to open-ended generation tasks (TriviaQA, ToxiGen, and TruthfulQA). Our experimental results reveal that SADI significantly outperforms existing activation intervention methods.

Our contributions are summarized as follows:

- We propose a dynamic activation intervention approach named Semantics-Adaptive Dynamic Intervention (SADI), which automatically modulates LLM activations at inference time to adapt to varied input semantics without requiring any additional training (see Section 3).

- SADI is a generic steering method applicable to a wide range of LLMs. Through extensive experiments with four model backbones over eleven diverse tasks, SADI has proven to significantly enhance model performance, surpassing baseline methods by substantial margins, with accuracy improvements reaching up to +14.69 (see Section 4). Our detailed analysis demonstrates that interventions targeting attention heads (SADI-HEAD) consistently yields significant performance improvements across various tasks, validating the effectiveness of our dynamic steering approach (see Section 4).

- SADI demonstrates excellent generalizability across different model sizes, few-shot settings, and multilingual scenarios (see Section 5). We further show that SADI is a cost-effective steering method that necessitates only a small number of additional examples (i.e., 150 items) in developing a dynamic steering vector and does not require any training (see Section 6).

## 2 RELATED WORK

Activation engineering has been proposed as a cost-effective method to modify an LLM's activations during decoding (Hernandez et al., 2023; Zou et al., 2023; Wang et al., 2024c). By analyzing

activation differences between contrastive pairs, these methods identify specific directions to adjust LLM behaviors. For instance, some studies have improved LLM truthfulness by shifting activations along vectors between true and false output distributions (Li et al., 2023b; Chen et al., 2024). Moreover, discrepancies in contextual examples have been used to identify crucial modifications needed to reduce LLM toxicity (Liu et al., 2023). More generally, these differences can be used to update the residual stream without requiring explicit direction settings for adjustments. Turner et al. (2023) construct steering vectors by assessing intermediate activation differences between two prompts, effectively shifting emotions from negative to positive. Similarly, Rimsky et al. (2023) use contrast pairs to create steering vectors that modify model behaviors by adjusting hidden states.

Our problem formulation aligns with the "linear representation hypothesis" (Park et al., 2024; Turner et al., 2023) which posits that high-level concepts (i.e., features of the input) are represented linearly as directions in the representation space. Model behavior intervention can be achieved by adding an appropriate steering vector to the representation of a concept without altering other concepts. One key insight is that intervening along the direction of the feature representation of an input is expected to enhance the probability of producing a desirable output. Intuitively, we need to preserve the semantic direction of an input when applying a steering vector to update activations.

Existing methods use fixed steering vectors generated from additional contrastive pairs during intervention without aligning with input semantics (Turner et al., 2023; Rimsky et al., 2023). As activation patterns can vary significantly across different inputs for the same task (Wang et al., 2024a;b), model steering with a fixed vector may cause the intervention direction to deviate from the representation of input contexts. This highlights the need for more adaptive approaches. Developing a dynamic steering mechanism to adapt to the input semantics is critical for effective intervention.

## 3 SEMANTICS-ADAPTIVE DYNAMIC INTERVENTION

In this section, we provide a comprehensive description of the proposed SADI. We begin with an overview of SADI in Section 3.1, and then introduce each step of SADI, including Difference Extraction (Section 3.2), Binary Masking (Section 3.3), and Adaptive Steering (Section 3.4).

### 3.1 OVERVIEW OF SADI

Our method, SADI, encompasses three pivotal steps to dynamically steer model behavior, as shown in Figure 1 and Algorithm 1. First, the activation differences between positive and negative examples are extracted across all layers of the model. These differences are aggregated to compute the mean difference, which is used to identify critical elements influencing the model's behavior. Based on this computation, we create an identification mask through binarization, keeping the crucial elements while masking out the insignificant ones. Furthermore, this mask is applied to the activations of user inputs, scaled by a factor during inference. In this way, we manage to manipulate the behaviors of LLMs. We present more details in the following sections.

### 3.2 DIFFERENCE EXTRACTION

In the initial step of SADI, our objective is to extract activation differences from contrastive pairs to isolate the internal activations most closely associated with the target behavior of the language model. Specifically, we aim to identify features that distinguish positive outcomes (e.g., correct answers) from negative ones (e.g., incorrect answers), thereby allowing adjustments to the model's behavior.

Let $\mathcal{P} = \{\mathcal{P}^l \mid 0 \leq l < L\}$ represents an LLM consisting of $L$ layers, whose behavior we seek to modify. We build a dataset $T = \{(x_i, y_i^{\text{pos}}, y_i^{\text{neg}})\}_{i=1}^N$ containing $N$ instances, where each instance includes the question $x_i$, a positive output $y_i^{\text{pos}}$, and a negative output $y_i^{\text{neg}}$. For each instance $i$ and layer $(l)$, we obtain activations by forwarding the concatenation of the input and the corresponding output through the model $\mathcal{P}$. Specifically, we forward $x_i$ concatenated with $y_i^{\text{pos}}$ to derive the positive activation $\boldsymbol{\mathcal{A}}_{i,j}^{\text{pos},(l)} = \mathcal{P}^l \left(x_i || y_i^{\text{pos}}\right)_j$, where $\cdot || \cdot$ denotes concatenation and $j$ represents the $j$-th token in the sequence. Similarly, we obtain $\boldsymbol{\mathcal{A}}_{i,j}^{\text{neg},(l)}$ for the negative activation. Additionally, we focus on the activation of the last token in each sequence, as it typically encapsulates the complete semantics

of the input-output pair. For simplicity, we denote the activation of the last token as $\mathcal{A}_i^{\text{pos},(l)}$ and $\mathcal{A}_i^{\text{neg},(l)}$ for the positive and negative outputs, respectively.

To identify the features within the model that differentiate correct from incorrect outputs, we compute the difference between the positive and negative activations for each instance at each layer as follows:

$$D_i^{(l)} = \mathcal{A}_i^{\text{pos},(l)} - \mathcal{A}_i^{\text{neg},(l)}. \tag{1}$$

By examining differences $D_i^{(l)}$, we can determine which activations are crucial for model's behavior.

### 3.3 BINARY MASKING

We now present how we construct a mask to identify and focus interventions on the critical model elements that affect model's behavior. As shown in Figure 1 (Step 2), after extracting the activation differences from contrastive pairs, we compute the mean difference across all instances and layers, and concatenate them to build the overall mean difference vector $D$ for all model elements:

$$D = \text{Concat}(D^{(0)}, D^{(1)}, ..., D^{(L-1)}), \text{ where } D^{(l)} = \frac{1}{N} \sum_{i=1}^{N} D_i^{(l)}. \tag{2}$$

Here, $D \in \mathbb{R}^{L \times d_m}$ represents the concatenated mean activation differences across all $L$ layers, and $d_m$ denotes the dimensionality of the model components, which may correspond to hidden states, attention heads, or neurons in FFNs.

We then binarize the mean activation difference $D$ to create an identification mask $M \in \mathbb{R}^{L \times d_m}$. This is done by setting the entries corresponding to the top-K elements with the largest differences to 1 and the rest to 0:

$$M[l, m] = \begin{cases} 1 & (l, m) \in E_K \\ 0 & \text{otherwise} \end{cases}, \tag{3}$$

where $l$ indexes the layers, $m$ indexes the model elements within a layer, and $E_K$ is the set of indices of the top-K elements with the highest mean activation differences. This step ensures that SADI focuses on the most impactful elements contributing to the desired behavior, reducing unnecessary alterations to non-essential elements and enhancing the efficiency of the intervention.

### 3.4 ADAPTIVE STEERING

Previous activation intervention methods modify all activation elements indiscriminately, which can disrupt the model's overall behavior (Li et al., 2023b; Rimsky et al., 2023). To mitigate this issue, we perform a focused intervention on the top-K elements during inference. By leaving irrelevant activations intact, our intervention becomes less intrusive and preserves the model's non-target behaviors. With this design, SADI precisely adjusts activations to minimize disruption to the model's residual functionalities. This approach is visualized in Step 3 of Figure 1.

Unlike previous studies using the fixed steering vector to intervene models, and inspired by Park et al. (2024), we design a steering mechanism that considers the semantic direction of the input. The steering vector dynamically adapts to the input's semantic direction (see Equation 5), maintaining the effectiveness of the intervention without deviating from the intended semantics.

For a given user input $q$, we first extract the activations of the last token from each layer: $\mathcal{A}_q^{(l)}$ for $l = 0, 1, ..., L-1$. We then concatenate these activations to form a single vector:

$$\mathcal{A}_q = \text{Concat}(\mathcal{A}_q^{(0)}, \mathcal{A}_q^{(1)}, ..., \mathcal{A}_q^{(L-1)}). \tag{4}$$

Next, we apply the identification mask $M$ to these activations and update them using:

$$\mathcal{A}_q' = \mathcal{A}_q + \delta(\mathcal{A}_q \odot M). \tag{5}$$

Here, $\odot$ represents the element-wise product, and $\delta$ is a hyperparameter controlling the strength of the intervention along the input's semantic direction. By calculating the steering vector based on the activations of input $q$, the intervention dynamically aligns with the input's semantics. This approach maintains the direction of the activation projections, ensuring that the intervention remains semantically relevant and effective. Subsequently, we complete the altered forward pass with the updated activations $\mathcal{A}_q'$.

**Hyperparameters $K$ and $\delta$**  Our method introduces two key hyperparameters: $K \in \mathbb{N}^+$, specifying the number of top elements targeted during the intervention, and $\delta \in \mathbb{R}^+$, controlling the strength of the intervention. We perform a hyperparameter sweep to empirically determine their optimal values. Detailed analysis of the hyperparameter selection is provided in Section 4.3.

By selectively targeting the most impactful activation elements and adapting our intervention to the input's semantic direction, our method effectively steers the model toward the desired behavior.

---

**Algorithm 1:** SADI: Semantics-Adaptive Dynamic Intervention

---

**Input** : $T = \{(x_i^{\text{pos}}, x_i^{\text{neg}})\}_{i=1}^{N}$, a set of contrastive pairs; $U = \{u_j\}_{j=1}^{K}$, a test set; $\mathcal{P}$, a pre-trained LLM; $A$, a function to extract activations from $\mathcal{P}$;

**Output** : $O$, the modified outputs collection;

1 $\Delta\mathcal{A} \leftarrow \mathbf{0}$                // Initialize the mean difference
2 **for** $i=1$ to $N$ **do**       // Collect and compute the mean difference of activations
3    |    $\Delta\mathcal{A} \leftarrow \Delta\mathcal{A} + (A(\mathcal{P}(x_i^{\text{pos}})) - A(\mathcal{P}(x_i^{\text{neg}})))$
4 **end for**
5 $\Delta\mathcal{A} \leftarrow \frac{1}{N}\Delta\mathcal{A}$
6 $M \leftarrow \text{binarize}(\Delta\mathcal{A})$            // Create the identification mask
7 $O \leftarrow []$                  // Intervene generation
8 **for** $j=1$ to $K$ **do**
9    |    $\mathcal{A}_j \leftarrow A(P(u_j))$          // Extract activations for each input
10   |    $S_j \leftarrow M \odot \mathcal{A}_j$        // Apply mask to activations and update
11   |    $\mathcal{A}_j' \leftarrow \mathcal{A}_j + \delta \times S_j$
12   |    $O \leftarrow O \bigcup \{P(\mathcal{A}_j')\}$      // Complete the modified forward pass
13 **end for**
14 **return** $O$

---

# 4 EXPERIMENTS

We present our experimental setup (Section 4.1) , comparative methods (Section 4.2), main results (Section 4.3), and specific contributions of SADI's constitutes (Section 4.4) in this section.

## 4.1 EXPERIMENT SETTINGS

In this subsection, we describe the experimental settings for evaluating SADI. First, we outline the tasks and the evaluation metrics used to assess model performance. Following that, we detail the construction of contrastive pairs. Finally, we introduce the selected LLMs.

**Tasks and Evaluation Metrics**  We conduct experiments on the following two types of tasks: multiple-choice tasks and open-ended generation tasks. For the multiple-choice tasks, we use datasets: COPA (Gordon et al., 2012), StoryCloze (Mostafazadeh et al., 2016), NLI (Bowman et al., 2015), MMLU (Hendrycks et al., 2021), SST2 (Socher et al., 2013), SST5 (Socher et al., 2013), BoolQ (Clark et al., 2019), and Winogrande (Sakaguchi et al., 2020), with response formats ranging from 2-way to 5-way choices. Detailed descriptions of the datasets are provided in Appendix A.1. We measure performance across these tasks using accuracy.

For the open-ended generation tasks, we apply SADI on TriviaQA (Joshi et al., 2017), TruthfulQA (Lin et al., 2022), ToxiGen (Hartvigsen et al., 2022) datasets. The Exact Match (EM) metric assesses TriviaQA, while TruthfulQA is evaluated using multiple-choice accuracy (MC) and pre-trained judge models for truthfulness[2] and informativeness[3]. ToxiGen is evaluated with a HATEBERT classifier[4] to measure toxicity. Dataset sizes are detailed in Table 9 (Appendix A.2).

**Contrastive Pairs Construction**  For multiple-choice tasks, we generate positive prompts by concatenating questions with correct answers and generate negative prompts using a randomly chosen incorrect answer. For TriviaQA, a unique approach involves using a blank space as the

---

[2]https://huggingface.co/allenai/truthfulqa-truth-judge-llama2-7B

[3]https://huggingface.co/allenai/truthfulqa-info-judge-llama2-7B

[4]https://huggingface.co/tomh/toxigen_hatebert

Table 1: The overall results of seven multiple-choice tasks in a zero-shot setting, performed by LLAMA2-7B-CHAT. "SFT + SADI" indicates that SADI is applied to instruction fine-tuned models. A dash indicates that the training dataset is unavailable.

| Task | COPA | StoryCloze | NLI | MMLU | SST2 | BoolQ | Winogrande | AVG |
|---|---|---|---|---|---|---|---|---|
| BASELINE | 70.80 | 65.06 | 63.11 | 44.90 | 88.63 | 70.52 | 50.91 | 64.85 |
| ITI | 77.20 | 68.50 | 63.97 | 46.07 | 91.38 | 74.10 | 52.80 | 67.72 |
| CAA | 75.20 | 74.65 | 64.13 | 46.17 | 91.16 | 74.98 | 52.64 | 68.42 |
| **SADI** | | | | | | | | |
| ├ SADI-HIDDEN | 81.00 | 55.99 | 59.28 | 45.66 | 92.15 | **76.25** | 52.64 | 66.14 |
| ├ SADI-NEURON | **82.20** | 67.57 | 62.97 | 46.91 | 88.69 | 70.40 | 51.93 | 67.24 |
| └ SADI-HEAD | 78.80 | **79.75** | **64.21** | **48.23** | **92.20** | 74.35 | **53.04** | **70.08** |
| SFT | 93.20 | 96.49 | 90.07 | - | 96.70 | 88.75 | 78.37 | 90.59 |
| **SFT + SADI** | | | | | | | | |
| ├ SADI-HIDDEN | **94.90** | 96.49 | 90.07 | - | 96.76 | 88.91 | 78.37 | 90.91 |
| ├ SADI-NEURON | 94.80 | 96.55 | **90.36** | - | **96.92** | 88.45 | 78.45 | 90.92 |
| └ SADI-HEAD | 94.60 | **96.55** | 90.30 | - | 96.81 | **88.94** | **78.61** | **90.97** |

incorrect answer. In `TruthfulQA`, we utilize data from its multiple-choice format to identify crucial elements. For the `ToxiGen` task, we leverage the `RealToxicityPrompts` dataset (Gehman et al., 2020), selecting entries with a toxicity score exceeding 0.955 to serve as negative prompts. This helps us pinpoint elements contributing to toxic outputs.

**Target LLMs** We evaluate the performance of SADI in enhancing the baseline model (BASELINE) - an instruction-tuned LLM, LLAMA2-7B-CHAT (Touvron et al., 2023). To verify the generalizability of SADI across various model backbones, we include three additional LLMs: BLOOMZ-7B (Muennighoff et al., 2023), MISTRAL-7B (Jiang et al., 2023), FALCON-7B-INSTRUCT (Almazrouei et al., 2023). These models are selected based on their demonstrated efficacy across diverse linguistic tasks and their widespread use in the research community. Furthermore, we extend our experiments to other models within the BLOOMZ family, specifically BLOOMZ-560M, BLOOMZ-1B, and BLOOMZ-3B, exploring how SADI performs across different model sizes.

## 4.2 EXPERIMENTAL COMPARISONS

In addition to evaluating SADI, we compare it against several approaches:

**Supervised fine-tuning (SFT)** We finetune all model parameters using the training dataset for each task, as previous works suggest that this approach serves as an upper bound for supervised finetuning. Specifically, we employ the AdamW optimizer with a learning rate of $2 \times 10^{-6}$ and a batch size of 4, conducting the fine-tuning across three epochs on four NVIDIA A-100 GPUs (80G).

**Inference-Time Intervention (ITI)** We follow Li et al. (2023b) in using contrastive pairs to identify the top heads for intervention. We sweep the hyperparameters of the heads involved and the strength of intervention to optimize results.

**Contrastive Activation Addition (CAA)** Rimsky et al. (2023) use the mean difference in the model's activations at the position of the answer letter between all the positive and negative prompts to construct a fixed steering vector to shift activations.

**Our Approach (SADI)** We compare three different configurations of the SADI shift. SADI-HIDDEN applies SADI to the identified key hidden states across all layers. SADI-HEAD modifies activations from the outputs of all attention heads across all layers. SADI-NEURON is based on the outputs from each non-linear activation function in the FFN blocks across all layers.

## 4.3 EXPERIMENTAL RESULTS

**SADI Significantly improves multiple-choice task performance.** As illustrated in Table 1, SADI demonstrates superior performance compared to the BASELINE and other intervention methods across multiple-choice tasks. While SFT consistently outperforms other methods in tasks with available training data, such as COPA and StoryCloze, its high data resource demands limit its effectiveness in tasks with scarce data, like MMLU. Both ITI and CAA show notable improvements, highlighting

Table 3: The overall results of three open-ended generation tasks performed by LLaMA2-7b-chat. The results of TriviaQA are obtained in a zero-shot setting, and we use 5-shot in-context learning in the ToxiGen and TruthfulQA tasks. † denotes results reproduced from other authors.

| Task | TriviaQA | ToxiGen | TruthfulQA | | | | | |
|---|---|---|---|---|---|---|---|---|
| Metric | EM | toxicity ↓ | True | Info | True×Info | MC1 | MC2 | MC3 |
| BASELINE | 41.60 | 49.71 | 66.83 | 99.51 | 66.50 | 33.41 | 51.07 | 24.76 |
| SFT | - | - | - | - | - | 24.20† | - | - |
| ITI | 42.80 | 45.27 | - | - | - | 34.64† | 51.55† | 25.32† |
| CAA | 43.20 | 49.71 | 71.60 | 83.84 | 60.03 | 34.03 | 52.76 | 25.62 |
| **SADI** | | | | | | | | |
| ├ SADI-Hidden | 43.80 | 34.43 | 35.13 | 51.73 | 25.38 | 67.07 | 92.90 | 62.31 |
| ├ SADI-Neuron | 43.50 | **17.14** | 74.54 | 93.51 | 69.71 | 34.88 | 52.50 | 25.79 |
| └ SADI-Head | **44.00** | 34.50 | **77.72** | 98.53 | **76.58** | **35.90** | **54.65** | **26.99** |

the effectiveness of intervention-based methods. SADI, leveraging dynamic interventions, achieves optimal performance overall, significantly outperforming the BASELINE. In comparison to fixed vector-based intervention methods (ITI and CAA), our dynamic intervention SADI yields substantial gains across all tasks. Specifically, SADI exceeds ITI and CAA by margins of +11.25 and +5.10, respectively, and surpasses the BASELINE by a large margin of +14.69 in the StoryCloze task. Furthermore, SADI can enhance the performance of task-specific fine-tuned models (SFT +SADI), underscoring its practicality for precise and targeted interventions.

**Improvement varies across SADI configurations.** Performance gains from the three SADI configurations vary, but all outperform ITI and CAA across tasks. Notably, SADI-Head achieves the most significant improvements, enhancing average accuracy by up to +5.23. While SADI-Hidden and SADI-Neuron also demonstrate strong performance improvements in certain scenarios, such as 76.25 for BoolQ and

Table 2: Average weighted accuracy on all four broad disciplines for MMLU task with LLaMA2-7b-chat.

| Domain | Humanities | STEM | Social | Other |
|---|---|---|---|---|
| BASELINE | 46.68 | 34.26 | 54.62 | 49.71 |
| **SADI** | | | | |
| ├ SADI-Hidden | 48.99 | 35.03 | 55.82 | 48.51 |
| ├ SADI-Neuron | 48.83 | 37.46 | 55.74 | 50.65 |
| └ SADI-Head | **49.76** | **39.35** | **56.21** | **52.31** |

82.20 for COPA, they occasionally show slight decreases in specific tasks, like NLI. Nevertheless, manipulating attention heads consistently results in improvements across all tasks and achieves the highest scores in most cases. For a detailed analysis, we present the results covering various domains of knowledge of MMLU tasks, including humanities, STEM and social sciences and other are shown in Table 2. SADI consistently enhances performances across these domains compared to the BASELINE, with SADI-Head yielding the highest improvements, up to +5.09 in the STEM domain.

**SADI improves open-ended generation task performance.** We further evaluate the performance of SADI on the open-ended generation tasks in Table 3. SADI, in its three configurations, generally outperforms the BASELINE, except for SADI-Hidden in the TruthfulQA generation track. SADI-Hidden underperforms in the generation track, but shows significant improvements in the truthful multiple-choice track. This suggests that hidden states may be particularly sensitive to multiple-choice formats. Conversely, SADI-Head significantly boosts truthfulness, with improvements reaching up to +10.08 on the True×Info metric for TruthfulQA. This underscores the generalizability of SADI 's dynamic intervention, effectively tailoring activations to the semantics required by each task.

**Hyperparameters are task-specific.** In Figure 2, we sweep two hyperparameters to control the intervention: the number of identified key attention heads, and the strength of intervention. Results indicate that optimal settings for these hyperparameters markedly vary across different tasks. This variability underscores the importance of carefully balancing the number of heads engaged and the scale of their amplification. For precise task performance optimization, it is recommended to search optimal hyperparameters using data from the validation sets specific to each task.

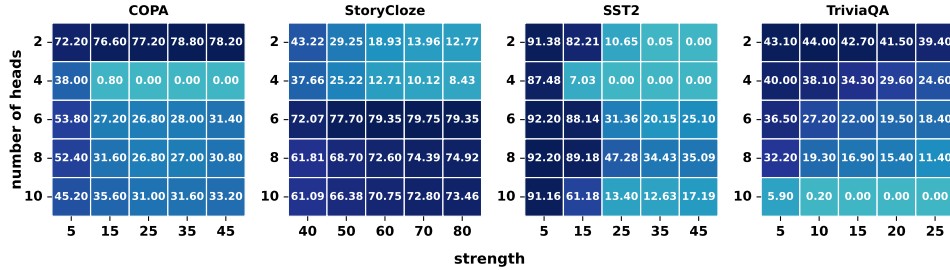

Figure 2: Results with varying intervention strength and numbers of key attention heads based on COPA, StoryCloze, SST2, TriviaQA tasks with LLaMA2-7B-CHAT.

## 4.4 SCRUTINIZE EFFECTS OF DYNAMIC INTERVENTION OF SADI

**SADI outperforms fixed steering and random intervention.** In Table 4, we conduct an ablation study to examine the contribution of Binary Masking (Step 2 in Section 3.3) and Adaptive Steering (Step 3 in Section 3.4) in SADI applied to tasks COPA, StoryCloze, Winogrande, SST2, and TriviaQA. SADI involves constructing an identification mask $M$, where we randomly assign K elements to 1 with the values of remainder elements set to 0 (termed "random identify" in Table 4). It can be observed from Table 4 that random element identification leads to a notable performance decrease, with reductions as great as -7.88 (dropping from 79.75 to 71.87 on StoryCloze). Subsequently, the mask $M$ is applied to the activations corresponding to the semantics of user inputs in SADI.

Additionally, we explore the effects of fixed steering, in which the mask $M$ is directly applied to the mean difference $D$, derived from the activations of contrastive pairs (see Eq. 2). If Step 3 (Eq. 5) employs fixed steering, it updates activations as:

$$\mathcal{A}'_q = \mathcal{A}_q + \delta(\boldsymbol{D} \odot M). \qquad (6)$$

The results in Table 4 show that using a fixed steering vector leads to significant performance degradation compared to the semantic-adaptive approach of SADI. This decline likely stems from a misalignment between the direction of intervention and that of input semantics.

Table 4: Ablation study for randomly identifying key elements and intervening with a fixed steering vector for SADI-HEAD with LLaMA2-7B-CHAT.

|  | **SADI** | random identify | fixed steering |
|---|---|---|---|
| COPA | **78.80** | 72.40 | 71.20 |
| StoryCloze | **79.75** | 71.87 | 69.42 |
| Winogrande | **53.04** | 52.88 | 52.56 |
| SST2 | **92.20** | 91.48 | 91.43 |
| TriviaQA | **44.00** | 42.90 | 41.50 |

## 5 DISCUSSION

In this section, we examine the generalizability across multiple LLMs (Section 5.1), different model sizes (Section 5.2), few-shot settings (Section 5.3), and multilingual scenarios (Section 5.4).

### 5.1 GENERALIZABILITY ACROSS MULTIPLE LLMS

Table 5: Generalizability evaluation of SADI given by BLOOMZ-7B, MISTRAL-7B, and FALCON-7B-INSTRUCT on the COPA, BoolQ and NLI tasks .

| Task | COPA | | | BoolQ | | | NLI | | |
|---|---|---|---|---|---|---|---|---|---|
| LLMs | BLOOMZ | MISTRAL | FALCON | BLOOMZ | MISTRAL | FALCON | BLOOMZ | MISTRAL | FALCON |
| BASELINE | 76.40 | 84.80 | 62.20 | 91.28 | 71.80 | 71.83 | 54.81 | 53.33 | 52.29 |
| SFT | 86.80 | 86.80 | 88.20 | 90.52 | 86.57 | 84.22 | 57.41 | 89.94 | 55.45 |
| **SADI** | | | | | | | | | |
| ├ SADI-HIDDEN | 76.40 | **93.00** | 60.60 | 91.31 | 49.43 | 62.73 | 53.25 | 35.69 | 43.67 |
| ├ SADI-NEURON | **82.20** | 84.20 | **62.40** | **91.40** | **74.61** | **74.48** | **56.43** | **54.07** | 52.48 |
| └ SADI-HEAD | **82.20** | 92.00 | 62.20 | 91.28 | 67.38 | 73.23 | 55.37 | 53.65 | **54.83** |

**SADI enhances performance across multiple LLMs in various tasks.** An important question is whether SADI can generalize over various LLMs. We apply SADI to three other well-performing

LLMs: BLOOMZ-7B, MISTRAL-7B, and FALCON-7B-INSTRUCT in COPA, BoolQ and NLI tasks. According to the results in Table 5, SADI consistently enhances performance across tasks compared to the BASELINE, despite varying improvement levels by configuration. It is noteworthy that the SADI-NEURON configuration with BLOOMZ-7B achieves the most substantial performance gains in all three tasks, demonstrating that different models may exhibit distinct functional elements.

## 5.2 GENERALIZABILITY ACROSS MODEL SIZES

**SADI outperforms SFT in smaller LLMs.** We further investigate the effectiveness of SADI on the BLOOMZ series across various model sizes. As shown in Table 6, SADI maintains its improvement over BASE-LINE even with smaller LLM sizes, confirming its generalizability across model sizes. Notably, SADI outperforms SFT with incremental gains of up to +1.2 for BLOOMZ-1B and +2.4 for BLOOMZ-560M, highlighting its effectiveness, especially in smaller models.

Table 6: Generalizability evaluation on BLOOMZ series in COPA task.

| Size | 7b | 3b | 1.1b | 560m |
|---|---|---|---|---|
| BASELINE | 70.8 | 79.2 | 49.8 | 50.0 |
| SFT | **88.8** | **85.8** | 50.6 | 52.0 |
| **SADI** | | | | |
| ├ SADI-HIDDEN | 76.4 | 79.2 | 50.0 | 50.0 |
| ├ SADI-NEURON | 74.0 | 79.2 | 50.0 | 52.6 |
| └ SADI-HEAD | 78.8 | 79.2 | **51.8** | **54.4** |

## 5.3 GENERALIZABILITY IN FEW-SHOT SETTINGS

**SADI improves few-shot performance but with less gains.** In Table 7, we compare SADI-HEAD to the BASELINE across zero-shot and few-shot settings on the SST5, Winogrande, and TruthfulQA tasks. The results highlight the generalizability of SADI in enhancing model performance with few-shot prompting across various tasks. While manipulating heads improves the performances in few-shot settings, the gains are less pronounced compared to those in zero-shot settings. This suggests that few-shot examples already provide a strong learning signal (to both BASELINE and SADI), somewhat overshadowing the additional benefits derived from head manipulation.

Table 7: Comparisons between few-shot and zero-shot on the SST5, Winogrande, and TruthfulQA.

| Task | SST5 | | Winogrande | | TruthfulQA MC1 | | TruthfulQA MC2 | | TruthfulQA MC3 | |
|---|---|---|---|---|---|---|---|---|---|---|
| Configuration | 0-shot | 5-shot | 0-shot | 5-shot | 0-shot | 5-shot | 0-shot | 5-shot | 0-shot | 5-shot |
| BASELINE | 28.24 | 53.21 | 50.91 | 52.01 | 27.66 | 33.41 | 44.45 | 51.07 | 20.59 | 24.76 |
| SADI-HEAD | 35.43 | 54.07 | 53.04 | 53.35 | 32.19 | 35.99 | 50.81 | 54.65 | 24.83 | 26.99 |

## 5.4 GENERALIZABILITY IN MULTILINGUAL SCENARIOS

Table 8: Evaluating SADI on multilingual task XCOPA with LLAMA2-7B-CHAT.

| Language | id | it | sw | ta | th | tr | vi | zh | AVG |
|---|---|---|---|---|---|---|---|---|---|
| BASELINE | 51.40 | 61.20 | 50.20 | 49.40 | 50.80 | 49.40 | 51.80 | 62.80 | 53.38 |
| **SADI** | | | | | | | | | |
| ├ SADI-HIDDEN | 51.40 | 62.40 | 50.00 | **50.00** | 51.20 | 48.80 | 52.20 | 64.80 | 53.85 |
| ├ SADI-NEURON | **63.60** | 68.80 | 50.20 | 48.80 | **53.80** | 50.60 | **60.40** | **70.40** | 58.33 |
| └ SADI-HEAD | 62.60 | **70.60** | **50.80** | 49.60 | 51.40 | **51.60** | 60.20 | 70.10 | **58.36** |

**SADI enhances performance in multilingual scenarios.** Although our primary experiments are in English, extending SADI to multilingual scenarios reveals its broader applicability. We further evaluate SADI on the multilingual XCOPA task (Ponti et al., 2020), covering eight languages: Indonesian (id), Italian (it), Swahili (sw), Tamil (ta), Thai (th), Turkish (tr), Vietnamese (vi), Chinese (zh). Table 8 illustrates varying degrees of performance enhancements across different languages. It can be observed that Indonesian shows the highest improvement, while Swahili gains the least. Despite these variations, SADI consistently boosts performance across diverse language settings and configurations. A detailed analysis of the key components effective language-wise is provided in Appendix A.3. Additional insights from cross-lingual evaluations are discussed in Appendix A.4.

## 6    ANALYSIS

In this section, we analyze activation difference distribution patterns for key model elements (Section 6.1), and how SADI behaves under varying numbers of contrastive pairs (Section 6.2).

### 6.1    CHARACTERISTICS OF ACTIVATION DIFFERENCE

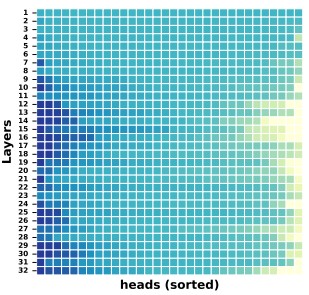

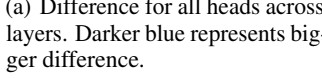

(a) Difference for all heads across layers. Darker blue represents bigger difference.

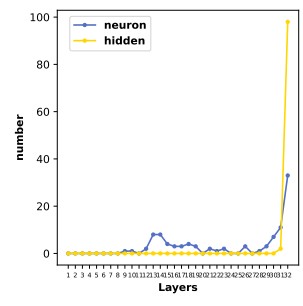

(b) Distribution of top-100 activation difference of neurons and hidden states. The y-axis represents the number of neurons or hidden states in top-100.

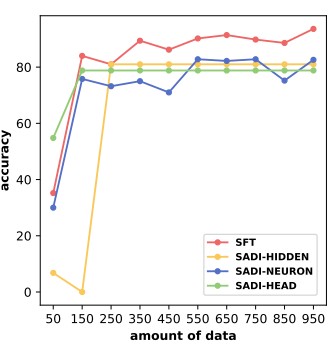

Figure 4: Relationship between accuracy and the amount of contrastive pairs.

Figure 3:  Activation difference of each head across layers and the distribution of top-100 activation difference of neurons and hidden states with LLaMA2-7B-CHAT in StoryCloze.

Figure 3(a) reveals activation patterns of attention heads in the middle to latter layers for contrastive pairs in StoryCloze, indicating primary information processing within these layers.  Figure 3(b) shows that activation differences in neuron activity and hidden states are concentrated in the latter layers, with the most significant discrepancies observed in the final layer. These consistent patterns across tasks (see Appendix A.5) lend support to the functional segregation hypothesis, which posits that latter layers are associated with language generation and middle layers are responsible for reasoning (Zhao et al., 2024). Given this, our interventions on attention heads likely influence both reasoning and generation, contributing to the consistent improvements.

### 6.2    SADI AND SFT WITH VARYING DATA

We assess the impact of varying amounts of contrastive pairs on SADI and SFT in COPA task. As shown in Figure 4, SFT performance improves with an increasing number of fine-tuning data. In contrast, SADI achieves optimal results with significantly fewer pairs, e.g., only 150 items are sufficient to calculate an identification mask for targeting critical heads for intervention. This highlights SADI's effectiveness and efficiency in low-resource conditions.

## 7    CONCLUSION

In this study, we propose Semantics-Adaptive Dynamic Intervention (SADI), a novel approach designed to dynamically steer model behavior by adapting to the semantic contexts of inputs. SADI enhances model adaptability by modulating the activations of the identified critical model elements during inference, taking into account the directions of input semantics. Our extensive experiments across various tasks, LLMs backbones, and languages settings have demonstrated that SADI significantly outperforms established baselines, offering generalizable improvements without requiring additional training.  Our study advances the field of "activation engineering" in LLMs, with the potential to inform the development of more advanced LLM intervention techniques.

ETHICS STATEMENT

This work presents Semantics-Adaptive Dynamic Intervention (SADI), a method designed to enhance the performance of large language models (LLMs) by dynamically adjusting their activations without additional training. We have conducted extensive experiments across diverse tasks to evaluate SADI's effectiveness, but we recognize that biases in the underlying models and datasets may still affect outcomes. We encourage practitioners to use SADI responsibly, with careful consideration of fairness, accountability, and transparency. No human subjects were involved in this research, and all experiments were conducted using publicly available models and datasets, adhering to their respective licenses and use policies.

REPRODUCIBILITY STATEMENT

We are committed to ensuring the reproducibility of our findings in this work. To facilitate this, we provide comprehensive details of our proposed Semantics-Adaptive Dynamic Intervention (SADI) method in Section 3 of the main paper, including the algorithms and mechanisms for dynamic steering vector generation and application to different model components. The experimental setups, including model configurations, datasets, and evaluation metrics, are thoroughly described in Section 4. We utilize publicly available models: LLAMA2-7B-CHAT, BLOOMZ-7B, MISTRAL-7B, and FALCON-7B-INSTRUCT, and detail any modifications or specific settings used during experimentation. All datasets employed in our evaluation are standard benchmarks, and we include references and links to these resources for accessibility. To further support reproducibility, we will release our code used for experiments upon publication, enabling other researchers to replicate our results and extend our work.

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

# A Appendix

## A.1 Description of Tasks

- **COPA** Each question consists of a premise and two alternatives, with the task being to choose the alternative that more plausibly has a causal relationship with the premise.
- **StoryCloze** Each question requires a model to choose the correct ending to a four-sentence story for evaluating story understanding and script learning.
- **NLI** It is a collection of sentence pairs manually labeled for balanced classification with the labels entailment, contradiction, and neutral.
- **MMLU** It is a benchmark designed to measure knowledge acquired during pretraining, covering 57 subjects across STEM, the humanities, the social sciences, and more.
- **SST2** and **SST5** They are datasets used for sentiment analysis with 2 labels (negative, positive) and 5 labels (negative, somewhat negative, neutral, somewhat positive, or positive), respectively.
- **BoolQ** It is a question answering dataset for yes/no questions where questions are naturally occurring.
- **Winogrande** It a fill-in-a-blank task with binary options, with the goal of choosing the right option for a given sentence which requires commonsense reasoning.
- **TriviaQA** It is a realistic text-based question answering dataset which includes question-answer pairs from documents collected from Wikipedia and the web.
- **TruthfulQA** It is a benchmark to measure whether a language model is truthful in generating answers to questions. The benchmark comprises 817 questions that span 38 categories, including health, law, finance and politics. TruthfulQA includes both multiple-choice and generation tracks. The performance of multiple-choice track is gauged using multiple-choice accuracy (MC). This metric is based on the conditional probabilities of candidate answers given the question, with a successful result counted when the truthful answer ranks first. In the generation track, we use two pre-trained judge models to evaluate the truthfulness and informativeness.
- **ToxiGen** It is a dataset that contains implicitly toxic and benign sentences mentioning 13 minority groups.
- **XCOPA** A multilingual dataset, translated from the English COPA, is used to evaluate the capacity of models to transfer commonsense reasoning across languages.

## A.2 Dataset Sizes

The size of datasets for each task are described in the Table 9.

Table 9: The number of data used for identifying key elements and testing for 11 tasks.

| Task | COPA | StoryCloze | NLI | MMLU | SST2 | SST5 | BoolQ | Winogrande | TriviaQA | ToxiGen | TruthfulQA |
|---|---|---|---|---|---|---|---|---|---|---|---|
| # identify | 500 | 360 | 2000 | 12178 | 1000 | 1000 | 1000 | 1000 | 1000 | 1000 | 817 |
| # testset | 500 | 1511 | 5000 | 12178 | 1821 | 2210 | 3270 | 1267 | 1000 | 1400 | 817 |

## A.3 Elements Overlap

In our analysis, detailed in Figure 5, we examine the overlap of identified key elements (attention heads, neurons, and hidden states) across various tasks. We observed minimal overlap across components between open-ended generation and multiple-choice tasks, particularly in attention heads and neurons. This indicates a high degree of functional specialization within these components. Unlike attention heads and neurons, hidden states demonstrate greater overlap, mainly due to their positioning in the final layer which more directly influences the model's output (as shown in Figure 4). The specialization observed among attention heads supports the notion that multi-head attention mechanisms evolve uniquely according to the task similarities. It is consistent with the finding in Li et al. (2023a).

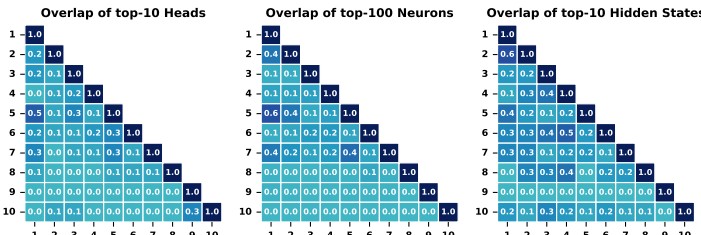

Figure 5: Overlap of identified key elements across various tasks. From 1 to 10 represents the tasks: COPA, StoryCloze, SST2, BoolQ, MMLU, NLI, Winogrande, TriviaQA, ToxiGen, TruthfulQA.

## A.4 CROSS-LINGUAL SADI

Table 10: Cross-lingual results based on the identified heads/neurons from the English contrastive pairs.

| Language | id | it | sw | ta | th | tr | vi | zh |
|---|---|---|---|---|---|---|---|---|
| BASELINE | 51.40 | 61.20 | 50.20 | 49.40 | 50.80 | 49.40 | 51.80 | 62.80 |
| SFT | 68.60 | 79.20 | 52.00 | 47.00 | 49.60 | 57.80 | 68.60 | 77.80 |
| SADI-HEAD | 62.80 | 67.60 | 50.40 | 50.20 | 51.00 | 51.20 | 56.60 | 70.00 |
| SADI-NEURON | 60.40 | 61.80 | 50.20 | 49.80 | 51.00 | 59.60 | 58.60 | 64.00 |

We have shown the effectiveness of SADI in the multilingual scenarios where contrastive pairs, constructed in the same language as the test input, are used to identify relevant components. Further, we explored its impact in a cross-lingual setting by employing English contrastive pairs to identify key elements and then applying SADI to multilingual test inputs. The results, along with those of SFT —which involves fine-tuning in English and testing on a multilingual dataset—are presented in Table 10. We found that SADI displayed enhanced language transfer capabilities, particularly in Tamil, Thai, and Turkish. These successful interventions suggest that critical elements are shared across languages, supporting SADI's utility in cross-lingual applications.

## A.5 ACTIVATION DIFFERENCE DISTRIBUTIONS ACROSS TASKS

We demonstrate the distributions of activation difference across layers for heads, neurons and hidden states in Figure 6 and in Figure 7 in COPA, BoolQ, TriviaQA, and TruthfulQA tasks. They show consistent patterns across various tasks.

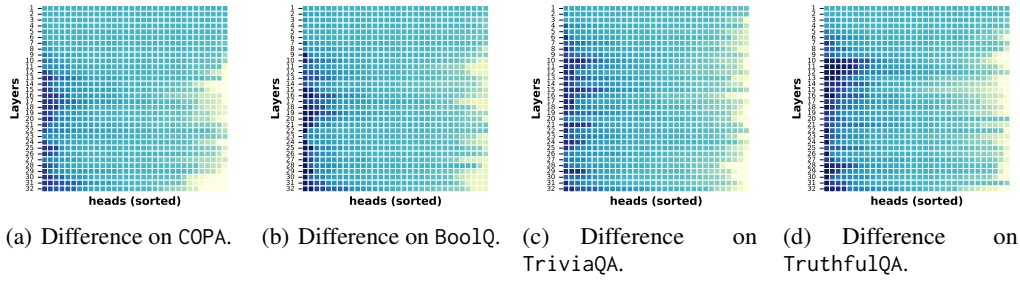

(a) Difference on COPA.    (b) Difference on BoolQ.    (c) Difference on TriviaQA.    (d) Difference on TruthfulQA.

Figure 6: Activation difference of each head across layer.

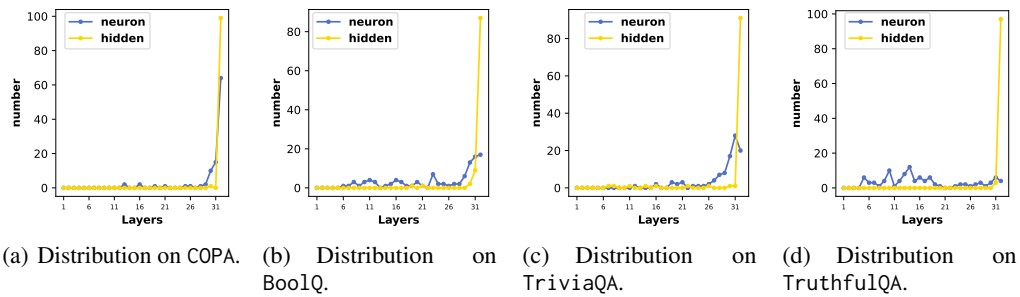

(a) Distribution on COPA.  (b) Distribution on BoolQ.  (c) Distribution on TriviaQA.  (d) Distribution on TruthfulQA.

Figure 7: Activation difference of top-100 neurons and hidden states.

