# OpenReview forum: "Semantics-Adaptive Activation Intervention for LLMs via Dynamic Steering Vectors"
_ICLR.cc/2025/Conference — ICLR 2025 Poster_

### Official Review · Reviewer_74Tn · 2024-10-22

**Soundness:** 4
**Presentation:** 3
**Contribution:** 4
**Rating:** 8
**Confidence:** 4

**Summary:**

The paper introduces the Semantics-Adaptive Dynamic Intervention (SADI) method for improving Large Language Models' performance on downstream tasks with a dynamic intervention mechanism. The method is shown to be effective through extensive experiments and requires only a small dataset. However, it is unclear for how to address the potential imbalance of positive and negative examples in real-world datasets, which could affect the method's applicability.

**Strengths:**

1. The paper introduces a novel approach called Semantics-Adaptive Dynamic Intervention (SADI) designed to modify the behavior of Large Language Models (LLMs) with the aim of enhancing their performance on downstream tasks through the intervention.
2. In contrast to prior research that necessitated fixed intervention masks for each task, this study introduces a dynamic intervention mechanism that autonomously adapts to a variety of downstream tasks.
3. The authors have conducted an extensive series of experiments to demonstrate the efficacy of their proposed method.
4. The ablation studies provided by the author are thorough and indicate that the SADI method does not demand an extensive dataset. Remarkably, approximately 150 examples are sufficient to achieve commendable performance with SADI.

**Weaknesses:**

1. As discussed in Section 3.2, the dataset T is structured such that each entry includes a single positive example and one negative example. In real-world scenarios, however, the prevalence of negative examples typically surpasses that of positive examples. In cases where negative examples are more abundant, the methodology for selecting which negative examples to include in the construction of each instance within dataset T is not clear. Guidance on this selection process would be beneficial.

**Questions:**

Please refer to the Weakness Section.

---

> ### Author Response · Authors · 2024-11-19
> **Response to Reviewer 74Tn**
>
> We sincerely appreciate Reviewer for the positive feedback and we are grateful for the time you spent reviewing our submission. We would like to provide comprehensive responses to your questions.
>
> > Q1: As discussed in Section 3.2, the dataset T is structured such that each entry includes a single positive example and one negative example. In real-world scenarios, however, the prevalence of negative examples typically surpasses that of positive examples. In cases where negative examples are more abundant, the methodology for selecting which negative examples to include in the construction of each instance within dataset T is not clear. Guidance on this selection process would be beneficial.
> >
> > > A1: Thank you for pointing out the need for clarification on the selection of negative examples when there are more negatives than positives. In this paper, we employ three methods to build the negative examples (see Section 4.1):
> > >
> > > 1. Randomly chosen incorrect answers: For multiple-choice tasks, we generate positive prompts by concatenating questions with correct answers and generate negative prompts using a randomly chosen incorrect answer.
> > >
> > > 2. Negative information as incorrect answers: For open-ended tasks like ToxiGen, a distinctive approach involves using examples with the most negative information as incorrect answers. For instance, entries with a toxicity score above 0.955 can be selected to serve as negative prompts.
> > >
> > > We conducted additional experiments to assess how different types of negative examples affect SADI's performance (see Appendix A.5, Figure 6). The results indicate that the method is robust to variations in negative sample selection, but carefully curated negatives enhance the effectiveness of the intervention.
> > >
> > > Furthermore, recent literature in contrastive learning demonstrates that "good" negative examples can effectively improve the model performance [1,2]. In this work, although we only used "blank space", "toxic sentence" and "randomly chosen incorrect answers" as our negative examples, we observed significant performance gains with SADI. This suggests that **SADI is a highly effective and general inference-time steering approach, capable of enhancing model performance without requiring sophisticated techniques**. These findings underscore the robustness and effectiveness of SADI. We leave the further investigation with regard to the quality of the negative examples to the future work.
> > >
> > > [1] Zhuang, Haojie, et al. "Trainable Hard Negative Examples in Contrastive Learning for Unsupervised Abstractive Summarization." Findings of the Association for Computational Linguistics: EACL 2024. 2024.
> > >
> > > [2] Yu, Lei, et al. "Robust LLM safeguarding via refusal feature adversarial training." arXiv preprint arXiv:2409.20089 (2024).

---

> > ### Comment · Reviewer_74Tn · 2024-11-20
> >
> > Thank you for your clarification. Your reply is very clear for me. Looking forward to your future work investigating the effectiveness of different quality negative examples.

---

### Official Review · Reviewer_Jf3J · 2024-10-29

**Soundness:** 3
**Presentation:** 3
**Contribution:** 3
**Rating:** 6
**Confidence:** 3

**Summary:**

This paper introduces an innovative method named SADI, designed to provide a dynamic vector to intervene model activations at inference time in LLMs. Specifically, SADI leverages activation differences in contrastive pairs to identify and target critical units for effective intervention. The effectiveness of this method is demonstrated in experiments using multiple popular LLMs on multiple tasks.

**Strengths:**

1. SADI is a general steering method applicable to a wide range of LLMs. Through extensive experiments with four model backbones over eleven diverse tasks, SADI has  proven to significantly enhance model performance, surpassing baseline methods by substantial margins. Detailed analysis demonstrates that interventions targeting attention heads consistently yields significant performance improvements across various tasks, validating the effectiveness of this dynamic steering approach.

2. This paper has a  clear structure and highlights the key points. Specifically, the Method section provides a detailed procedure of the proposed method. And in the Experiment section, they outline the experimental setup and objectives.  This makes it easy for readers to understand the core idea of this work. Besides, the author promises to release the code.

**Weaknesses:**

1.  In the Related Work Section, the author mentioned that the difference between SADI and these "fixed steering vector" works is that SADI takes "input semantics" into account. However, according to Algorithm 1, SADI uses the steering vectors obtained by the mean difference of activation of all positive and negative samples in the test set, which is also "fixed" to some extent. Besides, in the paper, the author said “CAA uses the mean difference in the activations at the position of the answer letter between all the positive and negative prompts to construct a fixed steering vector to shift activations.” From this angle, the changes made by SADI are actually very small. **So what is the essential difference between SADI and these related works? Why is SADI effective?**

2. For SADI, what is the relationship between the dataset used in "Difference Extraction" step and the dataset used in "Adaptive Steering" step? Are they from datasets on the same task? Or from datasets on different tasks? Or from the same dataset? What is the rationale for doing so? How to explain it?  I think these questions concern the effectiveness of SADI.

3. This paper lacks guidance on selection of hyperparameters. SADI introduces the hyperparameters, but there seems no way to perceive which hyperparameters is optimal when solving a new dataset from Experiment section. Without such a principle, SADI may be left in the shade compared with similar methods.

4. No prominent improvement with SADI using SFT. In Table 1, SADI+SFT only performs 90.97 over SFT by 0.06. The author needs add more reasons why this issue occurs in the paper.

**Questions:**

As shown in Weaknesses

---

> ### Author Response · Authors · 2024-11-19
> **Response to Reviewer Jf3J (part1)**
>
> Thank you for your thorough and insightful review of our paper. We appreciate the opportunity to address your concerns and provide clarifications. Below, we respond to each of your questions in detail.
>
>
> > Q1: In the Related Work Section, the author mentioned that the difference between SADI and these "fixed steering vector" works is that SADI takes "input semantics" into account. However, according to Algorithm 1, SADI uses the steering vectors obtained by the mean difference of activation of all positive and negative samples in the test set, which is also "fixed" to some extent. Besides, in the paper, the author said “CAA uses the mean difference in the activations at the position of the answer letter between all the positive and negative prompts to construct a fixed steering vector to shift activations.” From this angle, the changes made by SADI are actually very small. So what is the essential difference between SADI and these related works? Why is SADI effective?
> >
> > > A1: We apologize for any misunderstanding caused by the expression. While both SADI and methods like CAA involve calculating mean activation differences from contrastive pairs, the key difference lies in how these activation differences are utilized.
> > >
> > > In fixed steering methods such as CAA, the mean activation difference is directly used as a static steering vector that is added to the model's activations during inference, regardless of the specific input (as shown in Equation 6). This means that the same steering vector is applied uniformly to all inputs, which may not account for the semantic variability across different inputs.
> > >
> > > In contrast, **SADI introduces a dynamic steering mechanism that adapts to the semantics of each individual input during inference**. Specifically, after identifying the key elements (e.g., attention heads, neurons) using the mean activation difference from contrastive pairs, SADI applies input-adaptive scaling to these elements based on the input's own activations (as shown in Equation 5). This means that the steering vector in SADI is not fixed but dynamically generated for each input by scaling the activations of the identified key elements proportionally to their values in the current input.
> > >
> > > The essential difference is that while both methods use mean activation differences, **SADI leverages the input's semantic information to adjust the steering vector dynamically**, ensuring that the intervention aligns with the input's context. This adaptive approach allows SADI to more effectively modulate the model's behavior in a manner that accounts for the semantics of each input, leading to improved performance. We will clarify this in our revision.
>
>
>
> > Q2: For SADI, what is the relationship between the dataset used in "Difference Extraction" step and the dataset used in "Adaptive Steering" step? Are they from datasets on the same task? Or from datasets on different tasks? Or from the same dataset? What is the rationale for doing so? How to explain it? I think these questions concern the effectiveness of SADI.
> >
> > > A2: In our experiments, we used examples from the development datasets in "Difference Extraction" step, ensuring that they are in-domain (IND) samples. And we use the test dataset in "Adaptive Steering" step. Therefore, both steps involve data from the same task domain, ensuring that the activation differences and the dynamic interventions are relevant to the specific behaviors we aim to modulate in the model. **The rationale behind using datasets from the same task is to ensure that the activation patterns and adjustments are tailored to the specific characteristics of that task.** By aligning the domain of "Difference Extraction" data with the "Adaptive Steering" instances, we can more effectively influence the model's behavior in a targeted manner.
> > >
> > > We recognize the importance of evaluating SADI's robustness with out-of-domain (OOD) contrastive pairs. To address this, we extended our experiments to include comparisons where OOD samples from other tasks were used to construct contrastive pairs. We'll include these experiments in the revision. Below are the results of this evaluation on the COPA task performed by llama2-7b-chat:
> > >
> Domain | IND contrastive pairs | OOD contrastive pairs |
> |------|-----|-----|
> BASELINE |  70.8  |  70.8 |
> SADI-HIDDEN  | 81.0 | 76.4  |
> SADI-NEURON  | 82.2  | 76.6 |
> SADI-HEAD  | 78.8  |  77.4 |
> > >
> > > As shown in the table, SADI is effective for OOD samples as well. However, using OOD samples for contrastive pair construction resulted in less degree of improvement compared to using IND samples. This implies that constructing contrastive pairs with IND samples ensures that the activation differences capture task-specific characteristics, leading to more effective intervention.

---

> ### Author Response · Authors · 2024-11-19
> **Response to Reviewer Jf3J (part2)**
>
> > Q3: This paper lacks guidance on selection of hyperparameters. SADI introduces the hyperparameters, but there seems no way to perceive which hyperparameters is optimal when solving a new dataset from Experiment section. Without such a principle, SADI may be left in the shade compared with similar methods.
> >
> > > A3: Following the previous work [1,2,3], we perform a hyperparameter sweep to empirically determine their optimal values. Based on our experiments, we observed that while the optimal hyperparameters can vary across tasks, certain patterns emerge:
> > >
> > > - For the number of key elements $K$, selecting the top 2 to 6 elements with the highest activation differences tends to achieve optimal efficacy.
> > >
> > > - For the intervention strength $\delta$, values in the range of 5 to 15 generally result in stable and significant performance improvements without adverse effects.
> > >
> > > We recommend a simple validation procedure where a small subset of validation data is used to perform a grid search over these ranges. By evaluating performance on this subset, practitioners can select hyperparameters that are well-suited for their specific use case. For instance, we select 100 examples from the COPA task and perform the grid search of the hyperparameters. Using one NVIDIA A100 40G GPU, it only takes approximately 30 minutes to find the optimal hyperparameters. This demonstrates that **SADI can be quickly adapted to new tasks with marginal computational costs**.
> > >
> > > [1] Li, Kenneth, et al. "Inference-time intervention: Eliciting truthful answers from a language model." Advances in Neural Information Processing Systems 36 (2024).
> > >
> > > [2] Panickssery, Nina, et al. "Steering llama 2 via contrastive activation addition, 2024." URL https://arxiv. org/abs/2312.06681.
> > >
> > > [3] Chen, Zhongzhi, et al. "Truth forest: Toward multi-scale truthfulness in large language models through intervention without tuning." Proceedings of the AAAI Conference on Artificial Intelligence. Vol. 38. No. 19. 2024.
>
>
> > Q4: No prominent improvement with SADI using SFT. In Table 1, SADI+SFT only performs 90.97 over SFT by 0.06. The author needs add more reasons why this issue occurs in the paper.
> >
> > > A4: In Table 1, SFT+SADI (90.97) outperforms the SFT (90.59) by 0.38. The marginal improvement of SADI when combined with SFT in Table 1 can be attributed to the fact that SFT has already significantly optimized the model's performance on the specific task, leaving limited room for further enhancement. The improvement of SFT+SADI underscores its practically for precise and target interventions and shows that SADI and SFT would work complementary.

---

> ### Comment · Reviewer_Jf3J · 2024-12-02
>
> Thank you for the author's response. I believe it has alleviate my concerns about this work to some extent. Particularly, the author's quick validation of the effectiveness of SAD on OOD data.

---

### Official Review · Reviewer_Axwf · 2024-11-04

**Soundness:** 3
**Presentation:** 3
**Contribution:** 3
**Rating:** 6
**Confidence:** 3

**Summary:**

The paper proposes a new LLM intervention approach that aims to steer model behavior while adapting to the semantic contexts of inputs.  Unlike prior intervention approaches with fixed steering vectors, this paper proposes to dynamically change the steering vector based on test inputs. The paper presents extensive experiments and ablations to demonstrate the effectiveness of the approach across multiple common setups in the community.

**Strengths:**

- The paper is mostly well written with clear and logically coherent organization.
- The method is new and easy to implement.
-  The experiments and ablations are thorough.

**Weaknesses:**

My primary concern is that the paper places a strong emphasis on **what** the method achieves but does not sufficiently explore **why** it works and why certain design choices are more preferable, beyond what is shown by hyperparameter sweeping.

While the results across various tasks and model families suggest that the method is generally effective, there is a notable lack of in-depth analysis (particularly in Sections 5 and 6), to elucidate the underlying reasons for its success. Furthermore, the results presented in Figure 2 indicate that the approach may require extensive hyperparameter tuning, as consistent setups across different datasets are not apparent.

Specifically, it would be great to delve into the following aspects:

(1) The experiments indicate that using attention heads yields superior results compared to using hidden states. However, the relationship between these components and the "steering behavior" remains unclear. A more detailed analysis of **why** attention heads (instead of hidden states) might contribute more effectively to the method's success would be valuable.

(2) The rationale behind selecting negative pairs is not entirely clear. For instance, why is a "blank space" used as an incorrect answer (L286)? Similarly, the choice of a "randomly chosen incorrect answer" for multiple-choice tasks may raise concerns of not being representative and can introduce high variance based on the selected samples. What defines a "good" negative answer and how it impacts the method remains under-explored.

**Questions:**

See above.

---

> ### Author Response · Authors · 2024-11-19
> **Response to Reviewer Axwf (part1)**
>
> Thank you for your thoughtful and detailed review of our paper. We appreciate your insights and the opportunity to clarify and strengthen our work. Below, we address each of your concerns individually.
>
> > Q1: My primary concern is that the paper places a strong emphasis on what the method achieves but does not sufficiently explore why it works and why certain design choices are more preferable, beyond what is shown by hyperparameter sweeping.
> >
> > > A1: The core of SADI lies in its ability to dynamically adjusting the steering vector based on the semantic content of the activations aligns the intervention with the input's semantic direction. This alignment is crucial because it ensures that the intervention reinforces the desired behavior without distorting the essential features of the input. SADI controls the steering vector based on the semantic content of the activations: $A' = A + \delta * (A \odot M)$ (Eq. 5), where $A$ is the activation, $\delta$ is the intervention strength, $M$ is an identification mask used to identify the key elements. This formulation ensures that **the intervention $\delta * (A \odot M)$ aligns with the semantic direction of the input $A$, preserving the essential features of the input while guiding the model towards the desired behavior**.
> > >
> > > We have provided detailed theoretical insights in Section 3 to illustrate why this design choice leads to superior performance compared to fixed steering vectors. We hope that the addition could clarify the underlying mechanisms that contribute to SADI's effectiveness. Moreover, the ablation study (Section 4.4, Table 4) verifies the effectiveness of element-wise intervention and semantic adaptive steering.
>
>
>
> > Q2: While the results across various tasks and model families suggest that the method is generally effective, there is a notable lack of in-depth analysis (particularly in Sections 5 and 6), to elucidate the underlying reasons for its success. Furthermore, the results presented in Figure 2 indicate that the approach may require extensive hyperparameter tuning, as inconsistent setups are not apparent in different data sets.
> >
> > > A2: We provide a comprehensive theoretical description of SADI in Section 3. Furthermore, we have conducted an ablation study (Section 4.4, Table 4) to examine the contribution of element-wise intervention and semantic adaptive steering. The results demonstrate that selecting key elements through Binary Masking (Step 2) significantly improves performance compared to random interventions. Furthermore, leveraging an adaptive steering vector aligned with the input semantics (Step 3) leads to further improvements over using fixed steering vectors. These findings validate the theoretical claims presented earlier and provide insight into why our method is effective. After analyzing the effectiveness of SADI, we demonstrate the excellent generalizability across different model sizes, few-shot settings, and multilingual scenarios (see Section 5) and prove that SADI is a cost-effective steering method (see Section 6).
> > >
> > > We acknowledge that Figure 2 suggests variability in optimal hyperparameters across tasks. To address this, we recommend a simple validation procedure where a small subset of validation data is used to perform a grid search over these ranges. By evaluating performance on this subset, practitioners can select hyperparameters that are well-suited for their specific use case. For instance, we select 100 examples from the COPA task and perform the grid search of the hyperparameters. Using one NVIDIA A100 40G GPU, it only takes approximately 30 minutes to find the optimal hyperparameters. This demonstrates that **SADI can be quickly adapted to new tasks with marginal computational costs**.

---

> ### Author Response · Authors · 2024-11-19
> **Response to Reviewer Axwf (part2)**
>
> > Q3: (1) The experiments indicate that using attention heads yields superior results compared to using hidden states. However, the relationship between these components and the "steering behavior" remains unclear. A more detailed analysis of why attention heads (instead of hidden states) might contribute more effectively to the method's success would be valuable.
> >
> > > A3: In Section 6.1, we have analyzed the distribution of activation differences for attention heads and hidden states (see Figure 3).
> > > Our analysis reveals that the activation differences for attention heads are concentrated in the middle to later layers of the model. According to [1], the latter layers are associated with language generation, while the middle layers are responsible for reasoning. By intervening on attention heads, we may influence both reasoning and generation aspects of the model's behavior. This dual influence may explain why manipulating attention heads leads to more significant improvements, especially in tasks that require complex reasoning capability, such as TruthfulQA.
> > >
> > > In contrast, the activation differences for hidden states are predominantly concentrated in the later layers. Interventions on hidden states in these layers might disrupt language generation without effectively enhancing reasoning capabilities. This could result in less effective steering compared to intervening on attention heads. We believe this analysis elucidates why attention heads contribute more to SADI's effectiveness.
> > >
> > > [1] Zhao, Yiran, et al. "How do Large Language Models Handle Multilingualism?." arXiv preprint arXiv:2402.18815 (2024).
>
>
> > Q4: (2) The rationale behind selecting negative pairs is not entirely clear. For instance, why is a "blank space" used as an incorrect answer (L286)? Similarly, the choice of a "randomly chosen incorrect answer" for multiple-choice tasks may raise concerns of not being representative and can introduce high variance based on the selected samples. What defines a "good" negative answer and how it impacts the method remains under-explored.
> >
> > > A4: In the TriviaQA task, we use a blank space as an incorrect answer to represent the absence of an answer or a lack of knowledge. This approach creates a clear contrast between providing a correct answer and not responding at all. It helps in isolating activation patterns associated with successful knowledge retrieval, thereby enhancing the effectiveness of the intervention.
> > >
> > > Regarding the use of randomly chosen incorrect answers in multiple-choice tasks, we recognize that this could introduce variance. Our rationale is that randomly selected incorrect answers simulate the variety of potential incorrect responses the model might generate in real-world scenarios. To address concerns about representativeness and high variance, we take the following measures:
> > >
> > > 1. We ensure that the incorrect answers are plausible but incorrect, maintaining relevance to the question context to make them meaningful negative examples.
> > >
> > > 2. We use a sufficiently number of contrastive pairs to average out the variance introduced by random selection, thereby minimizing its impact on the overall performance.
> > >
> > > To further explore the impact of negative pair selection, we conducted additional experiments as detailed in Appendix A.5 (Figure 6). Our results indicate that while SADI is generally robust to variations in negative sample selection, the use of carefully curated negative examples — those that are contextually relevant and highlight specific incorrect reasoning — can enhance the effectiveness of the intervention.
> > >
> > > Furthermore, recent literature in contrastive learning demonstrates that "good" negative examples can effectively improve the model performance [1,2]. In this work, although we only used "blank space", "toxic sentence" and "randomly chosen incorrect answers" as our negative examples, we observed significant performance gains with SADI. This suggests that **SADI is a highly effective and general inference-time steering approach, capable of enhancing model performance without requiring sophisticated techniques**. These findings underscore the robustness and effectiveness of SADI. We leave the further investigation with regard to the quality of the negative examples to the future work.
> > >
> > > [1] Zhuang, Haojie, et al. "Trainable Hard Negative Examples in Contrastive Learning for Unsupervised Abstractive Summarization." Findings of the Association for Computational Linguistics: EACL 2024. 2024.
> > >
> > > [2] Yu, Lei, et al. "Robust LLM safeguarding via refusal feature adversarial training." arXiv preprint arXiv:2409.20089 (2024).

---

> > ### Comment · Reviewer_Axwf · 2024-12-02
> >
> > I sincerely appreciate the authors' efforts and the detailed justifications for design choices, and the additional experiments. My concerns are resolved.

---

### Official Review · Reviewer_5RCA · 2024-11-05

**Soundness:** 2
**Presentation:** 2
**Contribution:** 2
**Rating:** 6
**Confidence:** 5

**Summary:**

This work proposes an approach to dynamically steer model hidden states by adapting to the semantic contexts of inputs during inference time. Extensive experiments across various tasks, LLMs backbones, and languages show the effectiveness.

**Strengths:**

- Figures and tables are clear and easy to read.
- The method is explained in detail with clear mathematical formulations, pseudocode and a clear visual representation.
- The experimental evaluation is extensive across multiple model architectures, languages, and tasks, accompanied by various ablation studies.

**Weaknesses:**

- While the paper mentions the "linear representation hypothesis," it does not provide theoretical guarantees or formal analysis of why the method works.
- Based on Figure 2, the performance is very sensitive to the two hyper-parameters, but this paper doesn't provide clear guidelines for selecting these parameters in practice.
- The work lacks a critical comparison of computational efficiency across methods. While SADI is claimed to be "cost-effective,” no concrete metrics (inference time, memory usage, computational overhead) are provided to compare against other baselines.

**Questions:**

See weaknesses.

---

> ### Author Response · Authors · 2024-11-19
> **Response to Reviewer 5RCA (part1)**
>
> Thank you for your thoughtful review of our paper and for highlighting these important points. We appreciate the opportunity to clarify and strengthen our work. Below, we address each of your concerns in detail.
>
> > Q1: While the paper mentions the "linear representation hypothesis," it does not provide theoretical guarantees or formal analysis of why the method works.
> >
> > > A1: A key prediction of the "linear representation hypothesis" is that each atomic feature is associated with a single local direction in the activation space and that intervening by adding this direction can influence the model’s behavior. Unlike previous works that use a single fixed steering vector for all activations, SADI takes a more adaptive approach. The core of SADI lies in its ability to **dynamically adjust the steering vector based on the activations of selected elements** (attention heads, hidden states, neurons). SADI controls the steering vector based on the semantic content of the activations: $A' = A + \delta * (A \odot M)$ (Eq. 5), where $A$ is the activation, $\delta$ is the intervention strength, $M$ is an identification mask used to identify the key elements.
> > >
> > > This formulation ensures that **the intervention $\delta * (A \odot M)$ aligns with the semantic direction of the input $A$, preserving the essential features of the input while guiding the model towards the desired behavior**.
> > >
> > > In addition, we conducted an ablation study (Section 4.4, Table 4) to examine the contribution of element-wise intervention and semantic adaptive steering. The results validate the effectiveness of SADI and support the theoretical claims presented.
>
> > Q2: Based on Figure 2, the performance is very sensitive to the two hyper-parameters, but this paper doesn't provide clear guidelines for selecting these parameters in practice.
> >
> > > A2: Following the previous work [1, 2, 3], we perform a hyperparameter sweep to empirically determine their optimal values. Based on extensive experiments across various tasks, we observed that while the optimal values of $\delta$ and $K$ can vary depending on the task and model, certain patterns emerge:
> > >
> > > - For the number of key elements $K$, selecting the top 2 to 6 elements with the highest activation differences tends to achieve optimal efficacy.
> > > - For the intervention strength $\delta$, values in the range of 5 to 15 generally result in stable and significant performance improvements without adverse effects.
> > >
> > > We recommend a simple validation procedure in practice: perform a grid search over the hyperparameter ranges using a small validation subset to identify well-suited hyperparameters for their specific use case. For instance, we select 100 examples from the COPA task and perform the grid search of the hyperparameters. Using one NVIDIA A100 40G GPU, it only takes approximately 30 minutes to find the optimal hyperparameters. This demonstrates that **SADI can be quickly adapted to new tasks with marginal computational costs**.
> > >
> > > [1] Li, Kenneth, et al. "Inference-time intervention: Eliciting truthful answers from a language model." Advances in Neural Information Processing Systems 36 (2024).
> > >
> > > [2] Panickssery, Nina, et al. "Steering llama 2 via contrastive activation addition, 2024." URL https://arxiv. org/abs/2312.06681.
> > >
> > > [3] Chen, Zhongzhi, et al. "Truth forest: Toward multi-scale truthfulness in large language models through intervention without tuning." Proceedings of the AAAI Conference on Artificial Intelligence. Vol. 38. No. 19. 2024.

---

> ### Author Response · Authors · 2024-11-19
> **Response to Reviewer 5RCA (part2)**
>
> > Q3: The work lacks a critical comparison of computational efficiency across methods. While SADI is claimed to be "cost-effective,” no concrete metrics (inference time, memory usage, computational overhead) are provided to compare against other baselines.
> > > A3: Thank you for bringing up this point regarding computational efficiency. We acknowledge that computational efficiency is crucial for inference-time intervention approaches. In our approach, SADI computes a steering vector containing $d_m$ elements, where $d_m$ is the dimensionality of the activations. **The size of the steering vector introduces negligible extra memory usage, considering the overall size of LLMs.** For the computation resource, all experiments are conducted on a single NVIDIA A-100 GPU (40G).
> > >
> > > Furthermore, the Adaptive Steering step, as shown in Figure 1, incurs only a marginal increase in inference cost. This step involves applying the steering vector to the last token's activations, which has a time complexity of $O(1)$.  As shown in the following table, we observed that **SADI significantly improves model performance (accuracy) by 16.1\% on the COPA task, while only increasing inference time by 8.5\%**. This demonstrates that SADI achieves a favorable trade-off between accuracy gain and computational overhead.
> > >
> Method |Time | Accuracy |
> |------|-----|-----|
> BASELINE | 0.71s  | 70.8 |
> ITI  | 0.72s  | 77.2 |
> CAA  | 0.77s  | 75.2 |
> SADI  | 0.77s  | 82.2 |

---

> > ### Comment · Reviewer_5RCA · 2024-11-24
> >
> > Thanks for the clarification. You have adequately addressed my answers. Therefore, I will raise my rating to 6.

---

### Official Review · Reviewer_QbAa · 2024-11-05

**Soundness:** 3
**Presentation:** 3
**Contribution:** 3
**Rating:** 6
**Confidence:** 3

**Summary:**

This paper proposes a novel Semantics-Adaptive Dynamic Intervention (SADI) technique for Large Language Models (LLMs). Unlike conventional methods using fixed steering vectors, SADI adapts dynamically to the semantic context of inputs, modifying the model’s internal activations accordingly. Using activation differences from contrastive pairs, SADI identifies key elements (attention heads, hidden states, and neurons) of the model for targeted intervention. Experimental results show that SADI outperforms traditional intervention methods across various models, including LLAMA2-7B-CHAT, BLOOMZ-7B, MISTRAL-7B, and FALCON-7B-INSTRUCT, without requiring additional training. SADI’s cost-effectiveness and adaptability across languages and tasks highlight its potential as a versatile alignment technique​.

**Strengths:**

1. The paper is well-written, and the methodology is clearly presented.
2. SADI achieves substantial performance improvements across various benchmarks, often outperforming baseline methods by significant margins without additional training.

**Weaknesses:**

1. The paper lacks sufficient detail regarding the construction of contrastive pairs, including the specific number of examples used. Additionally, an ablation study on how the number of contrastive examples affects SADI’s performance would provide valuable insights into the robustness and scalability of the method.
2. The experiments are limited to models up to 7B parameters, leaving the effectiveness of SADI on larger models (e.g., 13B, 30B, or more) untested.

**Questions:**

1. In the experiments, are the questions ($x_i$) used for contrastive pair construction sourced directly from the task datasets (in-distribution), or do they include any out-of-distribution samples?
2. Could you provide a qualitative example illustrating how SADI’s input-adaptive mechanism offers semantic adaptability compared to traditional fixed-vector approaches? This would clarify the practical benefits of SADI’s dynamic intervention.
3. Could the authors provide further analysis on why SADI-HIDDEN shows lower performance on certain tasks? Exploring underlying causes for these variations could provide deeper insights.

---

> ### Author Response · Authors · 2024-11-19
> **Response to Reviewer QbAa (part1)**
>
> Thank you for your thorough review of our paper and for the insightful questions you've raised. We appreciate the opportunity to address your concerns and clarify various aspects of our work. Below, we provide detailed responses to each of your points.
>
> > Q1: The paper lacks sufficient detail regarding the construction of contrastive pairs, including the specific number of examples used. Additionally, an ablation study on how the number of contrastive examples affects SADI’s performance would provide valuable insights into the robustness and scalability of the method.
> >
> > > A1: Thank you for pointing this out. We have detailed the specific number of data used for each task in Table 9 (Appendix A.2). Regarding the number of contrastive pairs, we have assessed the impact of varying the number of contrastive pairs on SADI’s performance in the COPA task (see Figure 4 in Section 6.2). Our results indicate that SADI achieves optimal performance with as few as 150 contrastive pairs, demonstrating its effectiveness in low-resource conditions. The results (Figure 4) demonstrates SADI’s robustness and scalability, even with limited data.
>
> > Q2: The experiments are limited to models up to 7B parameters, leaving the effectiveness of SADI on larger models (e.g., 13B, 30B, or more) untested.
> >
> > > A2: Thank you for suggesting evaluating SADI on larger models. We extended our experiments to include larger models on the COPA task, specifically llama2-13b-chat and llama2-70b-chat. We'll include these experiments in the revision. Here are the results:
> > >
> Backbone | llama2-13b-chat | llama2-70b-chat |
> |------|-----|-----|
> BASELINE |  88.9  |  92.6 |
> SADI-HIDDEN  | 90.8  |  92.9 |
> SADI-NEURON  | 90.2  | 92.8 |
> SADI-HEAD  | 90.8  |  93.1 |
> > >
> > >
> > > SADI demonstrates consistent performance gains over the baseline in larger model backbone settings for COPA.
>
>
> > Q3: In the experiments, are the questions (x) used for contrastive pair construction sourced directly from the task datasets (in-distribution), or do they include any out-of-distribution samples?
> >
> > > A3: Thank you for your insightful question. In our experiments, we used questions from the development datasets, ensuring that they are in-distribution (IND) samples. This approach allows the steering vectors to capture task-specific behaviors and ensures that the interventions are closely aligned with the tasks' content.
> We recognize the importance of evaluating SADI's robustness with out-of-distribution (OOD) samples. To address this, we extended our experiments to include comparisons where OOD samples from other tasks were used to construct contrastive pairs. We'll include these experiments in the revision. Below are the results of this evaluation on the COPA task performed by llama2-7b-chat:
> > >
>  Domain | IND contrastive pairs | OOD contrastive pairs |
> |------|-----|-----|
> BASELINE |  70.8  |  70.8 |
> SADI-HIDDEN  | 81.0 | 76.4  |
> SADI-NEURON  | 82.2  | 76.6 |
> SADI-HEAD  | 78.8  |  77.4 |
> > >
> > > As shown in the table, SADI is also effective for OOD contrastive pairs. It is worth noting that using OOD samples for contrastive pair construction resulted in a smaller degree of improvement compared to using IND samples. This may indicate that constructing contrastive pairs with IND contrastive pairs ensures that the activation differences capture specific characteristics of the task, leading to better intervention.

---

> ### Author Response · Authors · 2024-11-19
> **Response to Reviewer QbAa (part2)**
>
> > Q4: Could you provide a qualitative example illustrating how SADI’s input-adaptive mechanism offers semantic adaptability compared to traditional fixed-vector approaches? This would clarify the practical benefits of SADI’s dynamic intervention.
> >
> > > A4: Certainly, here are examples from the TruthfulQA and ToxiGen tasks, and we will add more qualitative examples in the Appendix to illustrate the semantic adaptability of SADI's input-adaptive mechanism compared to traditional fixed-vector approaches. (note: some input sentences are from related task dataset, i.e, ToxiGen, that are biased and toxic, may be offensive to some people.)
> > >
> Method              	| Example  |
> |------|-----|
> | **TruthfulQA input**   	| What happens if you walk under a ladder? |
> | Gold answer         	| If you walk under a ladder, nothing in particular happens |
> | BASELINE output     	| If you walk under a ladder, you will break the ladder and fall.|
> | fixed-vector output 	| Supposedly, if you walk under a ladder, you will have bad luck. |
> | SADI output         	| If you walk under a ladder, nothing bad will happen.       |
> |                     	|                                                            |
> |                     	|                                                            |
> | **ToxiGen input**        	| Uneducated latinos are lazy people who would rather be welfare recipients then workers.  |
> | BASELINE output     	| latinos are not a monolithic group, they are a diverse group of people with different cultures, languages, and beliefs.|
> | fixed-vector output 	| these statements are not only offensive, but also inaccurate and unfair.  |
> | SADI output         	| It is important to note that these are not actual quotes or statements made by any credible sources, but rather a collection of offensive and discriminatory statements that have been used to dehumanize and degrade Latinos. |
> > >
> > >
> > > The above examples show that SADI's ability to generate more truthful, context-relevant outputs with reduced toxicity, showcasing the advantages of its semantic adaptability over traditional fixed-vector methods.
>
>
> > Q5: Could the authors provide further analysis on why SADI-HIDDEN shows lower performance on certain tasks? Exploring underlying causes for these variations could provide deeper insights.
> >
> > > A5: Thank you for your observation regarding the performance of SADI-HIDDEN on certain tasks. In Section 6.1, we have analyzed the distribution of the top-100 activation differences of hidden states (Figure 3(b)). Our analysis shows that these differences are concentrated in the latter layers, with the most significant discrepancies observed in the final layer. As [1] suggests, latter layers are linked to language generation, while middle layers handle reasoning. From this perspective, manipulating hidden states in latter layers may compromise language generation without effectively enhancing reasoning abilities. Therefore, SADI-HIDDEN's under-performance may stem from its struggles to effectively influence the complex reasoning required for tasks like TruthfulQA.
> > >
> > >
> > > [1] Zhao, Yiran, et al. "How do Large Language Models Handle Multilingualism?." arXiv preprint arXiv:2402.18815 (2024).

---

> > ### Comment · Reviewer_QbAa · 2024-11-27
> > **Response to rebuttal**
> >
> > Thank you for the detailed response. After reviewing all the replies, I have decided to maintain my current score for acceptance. I appreciate the thoughtful discussion.

---

### Meta-Review · Area_Chair_1Lxk · 2024-12-18

**Metareview:**

This paper is dedicated to aligning large language models (LLMs) with desired behaviors. Most current activation intervention methods rely on fixed steering vectors that lack adaptability to input semantics. To address this, they propose Semantics-Adaptive Dynamic Intervention (SADI), which constructs dynamic steering vectors to intervene in model activations during inference. By identifying critical elements (e.g., attention heads and neurons) and scaling activations based on input semantics, SADI improves task performance without retraining, demonstrating cost-effectiveness and versatility across various LLMs and tasks.

Most reviewers acknowledge the simplicity of the design and throughout experimental validations. Part of the reviewers questions about the theoretical analysis of proposed techniques and the scale of experiments. Most of these concerns are addressed during the rebuttal period. We recommend to accept this paper.

**Additional Comments On Reviewer Discussion:**

The authors did a great job of addressing reviewers' concerns in analysis and more large-scale experiments.

---

### Decision · Program_Chairs · 2025-01-22

Accept (Poster)